# Proline Transport and Growth Changes in Proline Transport Mutants of *Staphylococcus aureus*

**DOI:** 10.3390/microorganisms10101888

**Published:** 2022-09-21

**Authors:** William R. Schwan

**Affiliations:** Department of Microbiology, University of Wisconsin-La Crosse, 1725 State St., La Crosse, WI 54601, USA; wschwan@uwlax.edu; Tel.: +1-(608)-785-6980

**Keywords:** proline, *Staphylococcus* *aureus*, transport, bacterial pathogenesis

## Abstract

*Staphylococcus aureus* is a major cause of skin/soft tissue infections and more serious infections in humans. The species usually requires the importation of proline to be able to survive. Previous work has shown that single mutations in genes that encode for proline transporters affect the ability of *S. aureus* to survive in vitro and in vivo. To better understand proline transport in *S. aureus*, double and triple gene mutant strains were created that targeted the *opuD*, *proP*, and *putP* genes. Single gene mutants had some effect on proline transport, whereas double mutants exhibited significantly lower proline transport. An *opuD prop putP* triple gene mutant displayed the lowest proline transport under low- and high-affinity conditions. To assess growth differences caused by the mutations, the same mutants were grown in brain heart infusion (BHI) broth and defined staphylococcal medium (DSM) with various concentrations of proline. The triple mutant did not grow in DSM with a low concentration of proline and grew poorly in both DSM with a high proline concentration and BHI broth. These results show that *S. aureus* has multiple mechanisms to import proline into the cell and knocking out three of the main proline transporters significantly hinders *S. aureus* growth.

## 1. Introduction

*Staphylococcus aureus* is the number one cause of skin/soft tissue infections in humans throughout the world, causing more than 700,000 of these infections each year in the United States alone [1,2]. Besides skin/soft tissue infections, *S. aureus* is a major cause of bloodstream infections with an incidence rate of 4.3–38.3 per 100,000 person-years in the United States [3] to 26.1 per 100,000 person-years among other industrialized countries [4]. Moreover, *S*. *aureus* is the most frequent cause of infective endocarditis in developed countries [5].

To flourish in the human body, most strains of *S. aureus* require proline be imported from the environment due to proline auxotrophy [6]. At least two proline transport systems have been characterized in *S. aureus*: a low-affinity system and a high-affinity system centered on PutP [7,8,9,10,11,12]. Within *S. aureus* and other bacterial species, proline serves as a source of carbon and nitrogen and as an osmoprotectant in high osmotic stress environments [13]. Several bacterial proline transporters facilitate the transport of proline into the bacterial cell. Within Gram-negative bacteria, there are at least three proline transporters able to transport proline into the bacterial cell: two low-affinity systems centered on ProP and ProU as well as the PutP high-affinity system [13] Similar proteins have also been identified for other Gram-positive bacteria, such as *Bacillus subtilis* [14,15].

The high-affinity proline transporter encoded by the *putP* gene is the most well-characterized proline transport system in *S. aureus* [10,12,16,17,18]. PutP has an important role in *S. aureus* survival in various animal tissues that have low proline concentrations [10,16,17]. Low proline concentrations and environments with high osmotic stress led to transcriptional activation of the *putP* gene in *S. aureus* growing both in vitro as well as within human abscesses and various murine tissues [18].

Low-affinity proline transport has been less studied in *S. aureus*. An *opuD* mutant transported less proline than the wild type strain when 400 μM of proline was used [19]. The *opuD* gene had the highest level of transcription in an environment with 174 μM of either proline or betaine [19,20,21]. A moderate to high osmotic stress environment also induced transcription of the *opuD* gene [19]. Although no significant difference was observed between the *opuD* mutant and wild type in a murine thigh abscess model, the *opuD* gene was regulated in *S. aureus* growing in murine abscesses.

To get a better assessment of proline transport in *S. aureus*, several proline transport system double mutants targeting the *opuD*, *proP*, and *putP* genes as well an *opuD*, *proP*, and *putP* triple mutant strain were constructed using transduction. These double and triple mutants were analyzed for proline uptake and growth in defined staphylococcal media (DSM) with various concentrations of proline in the broth media. We show that the triple mutant grew poorly in DSM and proline uptake was minimal, suggesting that proline transport is an essential facet of *S. aureus* survival during human infections.

## 2. Materials and Methods

### 2.1. Bacterial Strains, Media, and Growth Conditions

The bacterial strains used in this study are shown in Table 1. Strains SABS4-2, SABS6-1, SaBS7-1, SaBS8-1, SaBS10-12, and SaBS12-1 were generated using transductions described below. Brain heart infusion (BHI) broth and agar (Becton Dickinson, Sparks, MD, USA) were used to grow the *S. aureus* strains. DSM was used to propagate the *S. aureus* strains with different concentrations of proline added [18]. Proline concentrations (Sigma Aldritch, St. Louis, MO, USA) that were tested included 1×(1740 μM), 0.1× (174 μM), 0.01× (17.4 μM), and 0.001× (1.74 μM). For growth curves, the cultures were incubated in 20 mL of medium in 250-mL non-baffled Erlenmeyer flasks at 37 °C with shaking (250 rpm) without extra CO_2_ added. Sampling was done every two hours up to 12 h and then again after 24 h. Three separate growth curves analyses were done for each culture condition and strain. The media were supplemented with antibiotics (Sigma Aldritch, St. Louis, MO, USA) at the following concentrations: kanamycin 250 μg/mL, chloramphenicol 10 μg/mL, and erythromycin 5 μg/mL.

### 2.2. Transduction

To construct the double and triple proline transport mutants, transductions were performed. *S. aureus* strains 16F-157 (*putP* mutant) [10], 03856 (*proP* mutant) [22], and 06464 (*opuD* mutant) [22] were used as donors. All transductions were done with phage ϕ80α as described [24]. Transductants were selected on BHI agar containing erythromycin (5 μg/mL) or chloramphenicol (10 μg/mL) and 2 mM sodium citrate and incubated at 30 °C for 2 to 4 days.

### 2.3. Radioactive Proline Uptake Assay

All proline uptake assays were performed as described before [7,10] using 2.5 or 400 μM ^3^H-proline (Perkin-Elmer, Waltham, MA, USA). Briefly, *S. aureus* cells were grown to mid-logarithmic phase in 100 mL of BHI broth in 1 L non-baffled Erlenmeyer flasks with shaking (250 rpm) at 37 °C and pelleted by centrifugation at 6000× *g* for 5 min at 4 °C. Cells were washed twice with 25 mL of 50 mM potassium phosphate buffer (pH 7.5) and suspended in 12 mL of transport buffer (50 mM K_2_HPO_4_ [pH 7.5], 25 mM NaCl, 40 mM glucose) to a final concentration of approximately 50 μg total cellular protein/mL. Transport was assessed by the filtration method [7,10] where cells were pre-incubated at 37 °C for 5 min in the transport buffer and L-[2,3-^3^H)] proline was added at a final concentration of 2.5 μM or 400 μM and the bacterial cells shaken at 150 rpm at 37 °C. After 30 s, a 1 mL aliquot of each culture was filtered, the filter washed twice with unlabeled transport buffer, and then dried in scintillation vials. Samples were counted on a Beckman model LS 6000SC scintillation spectrophotometer (Beckman Coulter, Brea, CA, USA). The uptake assays were done at least two times per strain per condition tested. Wild type *S. aureus* RN4220 served as a positive control.

### 2.4. Statistics

A Student’s *t* test was used for statistical analyses using Microsoft Excel. *p* values of ≤0.05 were considered significant.

## 3. Results

### 3.1. Testing Proline Uptake in Single, Double and Triple Mutant Strains

Previously, proline uptake in strains with a single mutation in one of the proline transport systems was examined [10,19]. Through transduction, double and triple mutants in the OpuD, ProP, and PutP transport systems in *S. aureus* were generated. After the proline transport double and triple mutants were created, proline uptake was measured for each strain compared to the single mutants and wild type strain. To determine proline uptake into the various strains, two concentrations of proline were used: 2.5 μM (high- affinity) and 400 μM (low-affinity). Wild type strain RN4220 displayed proline uptake of 48.6 nmol/mg protein under high-affinity conditions and 796 nmol/mg protein under low-affinity conditions (Table 2). The *putP* (strain RN4220 putP) and *opuD* (SaBS7-1) mutants displayed lower proline uptake values in media with low and high concentrations of proline versus wild type that were not significant in the high proline containing environment and was similar to previous studies [10,19]. However, proline uptake was significantly lower for the *putP* mutant in a low proline containing environment and matched was previously reported [10]. A *proP* mutant strain (SaBS4-2) displayed proline uptake lower than wild type in both a low- and high-affinity environment, but neither number was significant.

Mutants that affected two proline uptake systems in *S. aureus* were also tested. The *proP putP* (strain SaBS6-1) and *opuD putP* (Strain SaBS 14-1) double mutants had significantly lower proline uptake compared to wild type under low- and high-affinity conditions (low affinity *p* < 0.02, high affinity *p* < 0.003). A *proP opuD* double mutant (strain SaBS8-1) had lower proline uptake versus wild type in both the low- and high- affinity conditions, but the low-affinity result was the only significant one (*p* < 0.05). Testing of the *opuD proP putP* triple mutant strain (SaBS10-12) showed the most significant differences in proline uptake. An 11-fold significant decrease in proline uptake was observed when 2.5 μM of proline was used (*p* < 0.003) and a 7.8-fold significant decline in proline uptake was seen when 400 μM of proline was tested compared to wild type (*p* < 0.02). These results demonstrated that mutants that knocked out two or more proline transport systems in *S. aureus* significantly affected the ability of *S. aureus* to import proline into the cell.

### 3.2. Growth Differences Occur When Two or More Proline Transported Systems Are Mutated

The proline uptake assays done above showed mutations that affected two or three proline transport systems had a significant effect on proline transport into *S. aureus* cells. To determine if those proline uptake differences corresponded to growth changes, each of the strains were grown in BHI broth as well as in DSM broth with proline concentrations that ranged from 1× (1740 μM) to 0.001× (1.74 μM).

Following growth in BHI broth, most of the mutants had growth curves similar to the wild type strain (Figure 1A). Strain SaBS8-1, the *opuD proP* double mutant, had significantly lower growth compared to wild type at all time points after 0 h (*p* < 0.05). The *opuD proP putP* triple mutant (SaBS10-12) grew significantly worse versus wild type for all of the time points after 0 h (*p* < 0.001).

When the strains were grown in 1× DSM (1740 μM) broth, most mutants exhibited growth that was similar to the wild type strain (Figure 1B). However, strain SaBS8-1 (*opuD proP*) showed significantly less growth than wild type for all of the time points (*p* < 0.05). Again, the triple mutant strain displayed the poorest growth that was significantly lower for the time points starting at 2 h when compared to the wild type strain (*p* < 0.001).

The inoculation of 0.1× DSM (174 μM) broth led to none of the single proline transport mutants displaying significantly lower growth compared to the wild type strain (Figure 1C). Each of the double mutant strains grew significantly less well after 2 h versus both the wild type and single mutant strains (*p* < 0.01). The *opuD putP* mutant strain showed the most attenuated growth of the double mutants that were tested. Again, the triple mutant displayed significantly poorer growth that was less than half of the growth observed for the wild type strain (*p* < 0.001).

An additional 10-fold decrease in the proline concentration to 17.4 μM in the 0.01× DSM broth did not cause a significant effect on the growth of the single mutants versus wild type (Figure 1D). Growth of the *proP putP* and *opuD proP* double mutants was attenuated versus wild type after 2 h (*p* < 0.05), whereas the *opuD putP* double mutant strain had even lower growth compared to wild type (*p* < 0.01). The triple mutant displayed the most significant growth attenuation effect after 2 h compared to wild type with a longer lag phase associated with the growth curve (*p* < 0.001).

Finally, the SaBS7-1 (*opuD*) and SaBS4-2 (*proP*) mutant strains showed growth curves comparable to wild type when grown in 0.001× DSM broth (Figure 1E). Growth of the *putP* mutant was significantly attenuated versus the wild type at all time points after 2 h and was similar to the *proP putP* double mutant growth curve (*p* < 0.05). The *opuD proP* double mutant only showed significant growth differences at time point after 2 h versus wild type (*p* < 0.05), and the *opuD putP* double mutant displayed growth that was slightly lower than the proP putP double mutant after 2 h (*p* < 0.01). No growth was observed for the triple mutant strain SaBS10-12. Thus, mutants with two or more proline transport systems that were knocked out had growth deficiencies in defined broth media with lower proline concentrations.

## 4. Discussion

Most strains of *S. aureus* require proline to be transported from the external environment to sustain their growth [6]. *S. aureus* is able to grow nearly anywhere within the human body, including within low proline containing environments. Proline concentrations within heart tissue vegetations [25], human urine [26], murine spleens [17], and murine livers [17] are quite low, so proline transport is vital for *S. aureus* cells to survive in these niches within a human. Previously, single mutations in the high-affinity proline transport gene (*putP*) [10,17] as well as a low-affinity transport gene (*opuD*) [19] were examined. Attenuated in vivo growth was also observed by the *putP* mutant in animal models of infection [10,16]). No studies have previously been done that examined double or triple mutations in proline transport systems in *Staphylococcus* and their effect on proline transport or growth.

Previous studies have shown that there are at least two proline transport systems in *S. aureus* [7,8,9,10,11]. In this study, it is shown for the first time that *S. aureus* has at least two functional low-affinity proline uptake systems, OpuD and ProP, in addition to the high affinity PutP system, which is similar to the three proline transport systems (ProP, ProU, and PutP) in *Escherichia coli* [13]. Mutants with a mutation in either the *opuD* or *proP* gene displayed modest proline uptake changes versus wild type, but no differences in their growth curves when the growth media contained 17.4 μM proline or higher concentrations. For other bacterial species, the ProP and OpuD transport systems have some redundancy in their presumed roles of bringing proline into the bacterial cells that are in an environment with higher proline concentrations (i.e., low-affinity) [13,27].

The only single mutant that had an effect on both proline uptake and growth was the *putP* mutant when it was grown in low proline containing conditions (i.e., high-affinity). This work matches previous work that showed proline uptake was significantly diminished when the *S. aureus* cells were grown in low proline media due to a *putP* mutation [10,12,17].

The most interesting findings from this study were observed when using double and triple proline transport system mutants. An *opuD proP* double mutant strain showed significant growth attenuation compared to wild type in all DSM media containing 17.4 μM or more of proline. Growth of SaBS8-1 (*opuD proP*) in BHI broth was also significantly less than wild type. By knocking out both low-affinity transport systems, *S. aureus* growth was hampered in high proline concentration environments. Both the *opuD putP* and *proP putP* mutants demonstrated worse growth in DSM containing less than or equal to 174 μM of proline compared to wild type. A proline transport double mutant (*putP proP*) in *E. coli* also showed an additive effect on survival compared to mutants with a single mutation [28].

Each of the double mutants had an effect on proline transport and growth. However, the *opuD proP putP* triple mutation had the most profound effect on proline uptake, showing barely any proline uptake in medium with a low concentration of proline (only 4.4 nmol/mg protein, high-affinity) as well as a 7.8-fold proline uptake decline in medium with a high concentration of proline (low-affinity). Not only was proline uptake crippled, but so was bacterial growth by the triple mutant in all of the media that was tested, including BHI broth. Strain SaBS10-2 (*opuD proP putP*) did not grow in DSM containing 1.74 μM proline and showed half or less growth in the other four types of media compared to wild type. A mutant with a *proP proU putP* triple mutation in *E. coli* also displayed significantly less proline uptake compared to wild type [29].

Why is proline transport so critical then for *S. aureus* survival? Besides providing proline as a carbon and nitrogen source [30], *S. aureus* also uses proline as an osmoprotectant. *Staphylococcus* is very osmotolerant, growing in salt conditions as high as 15% [29]. In terms of human health*, S. aureus* can overcome high salt conditions associated with the skin [31,32] and urinary tract [33,34]. As mentioned earlier, *S. aureus* is the leading cause of skin and soft tissue infections in humans that is responsible for nearly 700,000 of these infections each year in the United States [1]. Thus, the ability to be tolerant of osmotic stresses has been identified as an important virulence factor for the species [35,36].

By producing at least three proline transport systems (OpuD, ProP, and PutP) like *E. coli* and other Gram-negative bacteria do [37,38], *S. aureus* would have a better opportunity to internalize enough proline for osmoprotection and energy. The presence of two low-affinity proline transport systems (OpuD and ProP), as well as a high-affinity proline transport system (PutP), may provide opportunities to import enough proline for the species to survive in a number of environments where proline concentrations vary. All three proline transport systems are well conserved among *S. aureus* strains [35]. A recent study showed there are likely over 40 genes encoding gene products that are important for osmotolerance in *S. aureus*, some of them involved in proline transport [39].

Besides the dedicated OpuD, ProP, and PutP proline transport systems, *S. aureus* has other transporters, such as oligopeptide and ABC transporters, that can be used to bring oligopeptides that have proline attached to other amino acids. Individual mutations of some of these other transport systems can also affect *S. aureus* virulence in animal models of infection [40,41]. In an environment with high levels of proline, these other transport systems appeared to provide enough proline to allow some growth of the *opuD proP putP* triple mutant tested in this study. However, even these other transporters were unable to provide sufficient proline for the triple mutant when it was inoculated into DSM with only 1.74 μM of proline.

## 5. Conclusions

Previous work has shown some differences in *S. aureus* proline uptake and bacterial growth using single mutants that affected an individual proline transport system. In this study, mutations in two or more proline transport genes had a significant effect on *S. aureus* proline transport and growth in proline depleted media. Further, a triple *opuD proP putP* mutant displayed poor proline uptake and did not grow in defined medium with a very low proline concentration. Thus, proline importation is critical for *S. aureus* survival.

## Figures and Tables

**Figure 1 microorganisms-10-01888-f001:**
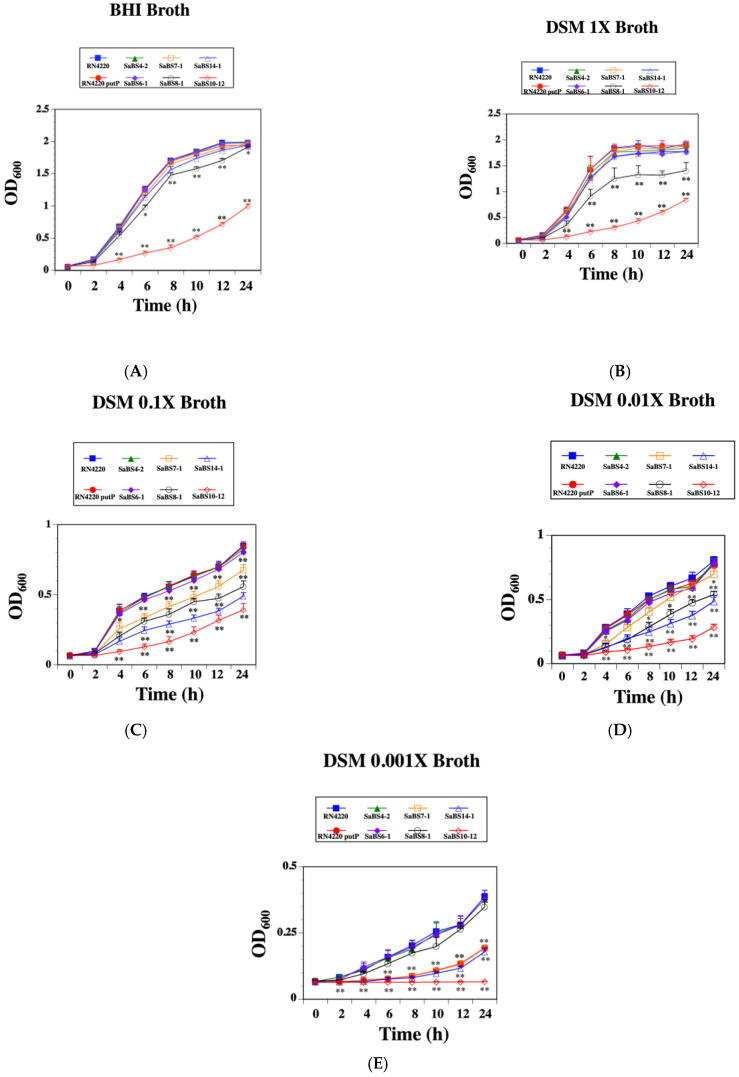
Growth curves of *S. aureus* RN4220 and derivatives in brain heart infusion broth (BHI) and defined staphylococcal minimal medium (DSM) with different proline concentrations. (**A**) BHI broth, (**B**) DSM 1× broth containing 1740 μM proline, (**C**), DSM 0.1× broth containing 174 μM proline, (**D**) DSM 0.01× broth containing 17.4 μM proline, and (**E**) DSM 0.001× broth containing 1.74 μM proline. The strains tested were wild type (closed blue square), RN4220 putP (*putP* mutant, closed red circle), SaBS4-1 (closed green triangle, *proP* mutant), SaBS7-1 (closed purple diamond, *opuD* mutant), SaBS6-1 (open orange square, *proP putP* mutant), SaBS8-1 (open black circle, *opuD proP*), SaBS14-1 (open blue triangle, *opuD putP* mutant), and SaBS10-12 (open red triangle, *opuD proP putP* mutant). The data represent the mean + standard deviation from at least three separate experiments. * *p* < 0.05 and ** *p* < 0.01.

**Table 1 microorganisms-10-01888-t001:** Bacterial strains used in the study.

Strain/Plasmid	Description	Reference
*S. aureus*		
3856	Newman *proP*::*mariner* (Em^R^)	[22]
6464	Newman *opuD*::*mariner* (Em^R^)	[22]
16F-157	RN6390 *putP*::Tn917	[10]
RN4220	Transformation-efficient *S. aureus* strain	[23]
RN4220 *putP*	RN4420 *putP*::Tn917	This study
SaBS4-2	RN4220 *proP*::mariner	This study
SaBS6-1	RN4220 *proP*::*mariner*, *putP*::Tn917	This study
SaBS7-1	RN4220 *opuD*::Km (Km^R^)	[19]
SaBS8-1	RN4220 *opuD*::Km, *proP*::*mariner*	This study
SaBS10-12	RN4220 *opuD*::Km, *proP*::mariner, *putP*::Tn917	This study
SaBS12-1	RN4220 *opuD*::Km, *putP*::Tn917	This study

**Table 2 microorganisms-10-01888-t002:** Radioactive proline uptake in *Staphylococcus aureus* proline transport mutant strains over a 30-s time course.

		Proline Uptake (nmol/mg Protein) ^a^	
Strains	Genotype	2.5 μM	400 μM
RN4220	Wild type	48.6 + 8.7 ^b^	796 ± 109
RN4220 *putP*	*putP*::Em Cm	18.4 + 1.0	572 ± 104
SaBS4-2	*proP*::Em	34.55 + 8.3	510 ± 149
SaBS6-1	*proP*::Em, *putP*::EmCm	13.7 + 1.7	163 ± 13
SaBS7-1	*opuD*::Km	37.1 + 6.1	505 ± 104
SaBS8-1	*opuD*::Km, *proP*::Tc	32.9 + 8.3	352 ± 88
SaBS10-12	*opuD*::Km, *proP*::Em, *putP*::Em	4.4 + 0.4	102 ± 26
SaBS14-1	*opuD*::Km, *putP*::EmCm	12.8 + 2.4	155 ± 18

^a^ Proline uptake measured after 30 s. ^b^ The data are the mean ± standard deviation for two to four separate runs.

## Data Availability

The data from this study are readily available from the author.

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
