# Peer review of "Proline Transport and Growth Changes in Proline Transport Mutants of Staphylococcus aureus"

_microorganisms, 2022, doi:10.3390/microorganisms10101888_

Round 1
Reviewer 1 Report
The manuscript by W. R. Schwan is a nice summary of the physiological consequences of proline transporter inactivation in Staphylococcus aureus. While the manuscript is data lean, the data presented are substantive and appropriate. It is easy to read and follow. A few things need clarification prior to publication.
1. Line 55 and elsewhere. Who is we? There is one author.
2. Line 70. In order for other researchers to reproduce the authors’ work, they must clearly state the cultivation conditions used in their experiments. This must include the flask-to-medium ratio, rpm of aeration, whether baffled flasks were used, and % CO2 if used.
3. Lines 145-148. I am surprised that growth in BHI produced such a low yield. Did you make dilutions of the bacterial cultures prior to taking the OD? Could the bacteria be growing microaerobically (see point 2.)?
Author Response
The manuscript has been rewritten. I want to thank the reviewers for their insightful comments on my manuscript # 1908655. My responses to each reviewer are noted below.
Reviewer #1
- References to we in the manuscript have been deleted.
- More information has been added in the materials and methods regarding flask-to-medium ratio, rpm of aeration, use of non-baffled flasks, and the absence of CO2.
- Dilutions were not made prior to OD readings for growth in BHI broth.

Reviewer 2 Report
The article "Proline Transport and Growth Changes in Proline Transport Mutants of Staphylococcus aureus " is a bit interesting. This study aims to show that mutations in two or more proline transport genes have a significant effect on S. aureus proline transport and growth in proline depleted media and demonstrate that proline importation is critical for S. aureus survival. However, there are some important weaknesses in this work.
First of all, the description of method part is too simple. The authors should describe how to construct the double and triple proline transport mutants in detail (Are they gene knock-out mutants or just point mutations?) and provide the primer list as the supplementary table and also depicted the radioactive proline uptake assay in detail such as how the S. aureus cells were treated before addition of proline. And the statistics part should be improved such as which software was used to do analysis.
Secondly, the figure is blurry. The authors should provide figures in high resolution. Also, there is no statistical significance shown on the figure?
Thirdly, I am wondering if the three proline transport systems are present in most S. aureus strains or just in RN4220?
At last, it would be more convincing if the author could construct the respective gene-complementing plasmid to verify that the effects on proline transport and bacterial survival were indeed caused by respective mutations in those proline transport-associated genes.
Therefore, I think the current work needs to be revised a lot in its current status.
Author Response
The manuscript has been rewritten. I want to thank the reviewers for their insightful comments on my manuscript # 1908655. My responses to each reviewer are noted below.
Reviewer #2
- The single mutation strains were previously made either by transposon mutagenesis or insertion of a Km cassette into the gene, which we noted in Table 1 with the appropriate references. Double and triple mutants were created by phage transduction. Wording has been added to the bacterial strains part of the manuscript to emphasize this. A more detailed description of the proline uptake assay has been added. All statistical analyses were done using Microsoft Excel, which has been added to the paper.
- The Figures have all been saved as high-resolution TIFF images and statistical significance has been added to each part of Figure 1.
- All three proline transport systems are well conserved in aureus. Language regarding this has been added to the discussion.
- The reviewer has a good point regarding complementation of the mutations. Past work has previously shown that complementation of the putP (Schwan et al., 1998) and opuD (Wetzel et al., 2011) genes restored proline uptake and virulence. Complementation of the proP gene has also been done and restored proline transport and survival, but was not reported in this paper. Complementation of the triple mutant would not be feasible.

Round 2
Reviewer 2 Report
I don't think the current resolution of figures in the manuscript is high enough for publication. The figures are still blurry to me. The authors could use other better graph software to improve them.
Author Response
The images have been reconfigured to have higher resolution.